# False-Positive Diagnosis of Congenital Heart Defects at First-Trimester Ultrasound: An Italian Multicentric Study

**DOI:** 10.3390/diagnostics14222543

**Published:** 2024-11-13

**Authors:** Silvia Andrietti, Serena D’Agostino, Marina Panarelli, Laura Sarno, Maria Laura Pisaturo, Ilaria Fantasia

**Affiliations:** 1Fetal and Perinatal Medicine Unit, IRCCS Policlinico San Martino, 16132 Genova, Italy; silvia.andrietti@gmail.com; 2Obstetrics & Gynaecology Department, San Giovanni di Dio Hospital, 88900 Crotone, Italy; dagostinoserena89@gmail.com; 3Fetal Medicine Unit, Di Venere Hospital, 70131 Bari, Italy; marina.panarelli@gmail.com; 4Department of Neurosciences, Reproductive Science and Dentistry, University Federico II, 80138 Naples, Italy; laurettasarno@gmail.com; 5Obstetrics & Gynaecology Unit, San Carlo Hospital, 85100 Potenza, Italy; marialaura.pisaturo@icloud.com; 6Obstetrics & Gynaecology Unit, San Salvatore Hospital, 67100 L’ Aquila, Italy

**Keywords:** congenital heart defects, first-trimester ultrasound, nuchal translucency, fetal echocardiography, ultrasound screening

## Abstract

**Objective.** Our objective was to assess the proportion of false-positive CHD cases at the first-trimester evaluation of the fetal heart, performed by experienced operators. **Methods.** This multicenter retrospective study included of pregnant women with suspicion of CHDs during first-trimester screening for aneuploidies. In all cases, the fetal heart assessments were performed by obstetricians with extensive experience in first-trimester scanning, following an extended protocol proposed by SIEOG national guidelines, which included an axial view of the fetal abdomen and chest to assess visceral situs and evaluation of the four-chamber view (4CV) and three-vessel trachea view (3VTV) with color Doppler. In all suspected cases, fetal echocardiography was offered within 16 and/or at 19–22 weeks’ gestation. **Results.** From a population of 4300 fetuses, 46 CHDs were suspected. Twenty-four cases were excluded from this analysis because the parents opted for early termination of the pregnancies due to associated structural and/or genetic anomalies. For the remaining 22, echocardiography was performed by 16 weeks in 14 cases (64%) and after 16 weeks in 8 cases. In 19 cases (86.4%), a fetal cardiologist confirmed the presence of a CHD. In three cases (13%), the cardiac anatomy was found to be normal at the fetal echocardiography and postnatally. **Conclusions.** This study shows that the proportion of false-positive cases at the first-trimester ultrasound examination of the fetal heart, performed by experienced operators, may carry a higher risk of false-positive diagnosis than expected. Therefore, this issue must be discussed in instances where a CHD is suspected at the first-trimester screening.

## 1. Introduction

Congenital heart defects (CHDs) affect approximately 4 to 13 of every 1000 live births and represent one of the leading causes of infant morbidity and mortality within the first year of life [1]. The early identification of CHDs through prenatal diagnosis has been shown to enhance pregnancy counseling and significantly improve neonatal outcomes by enabling timely planning for delivery and postnatal care [2,3,4]. Traditionally, fetal echocardiography (FE) conducted during the second and third trimesters is considered the gold standard for prenatal CHD detection due to its high accuracy and established protocols [5,6]. However, the widespread adoption of first-trimester screening, coupled with advancements in imaging technology and improvements in operator expertise, has facilitated earlier detection of congenital anomalies, including CHDs [7,8,9,10]. Recent studies have demonstrated that early fetal echocardiography performed by specialized fetal cardiologists before 16 weeks’ gestation can achieve a sensitivity of approximately 83.8% for detecting major CHDs, with a 74.5% concordance with follow-up echocardiograms conducted between 19 and 22 weeks of gestation [11,12,13]. These findings underscore the potential of early fetal cardiac assessment to identify significant heart defects well before mid-pregnancy. Research has indicated that the reliable detection of major CHDs during first-trimester ultrasounds is possible in both high-risk and low-risk populations, although the detection rates differ. Specifically, pooled sensitivity rates have been reported at 67.7% for high-risk populations and 55.8% for low-risk groups [14,15,16,17]. The factors influencing these detection rates include the operator’s experience, the characteristics of the study population, the specific types of cardiac abnormalities, and the application of a structured anatomical protocol [18]. Italian guidelines now recommend a detailed fetal anatomical assessment between 11 and 14 weeks of gestation [19,20], including a focused evaluation of the fetal heart, according to ISUOG (International Society of Ultrasound in Obstetrics and Gynecology) guidelines and particularly for high-risk pregnancies [21]. Notably, some Italian referral centers have extended this evaluation protocol to low-risk populations as well. The early detection of CHDs offers several potential benefits. For high-risk women, a normal first-trimester scan can provide significant reassurance, while abnormal findings can facilitate early counseling and planning. However, false positives present a considerable challenge, as a preliminary diagnosis of a CHD—later disproven upon follow-up—can cause substantial parental anxiety, increase the risk of unnecessary interventions, and contribute to the over-medicalization of pregnancies. The aim of this multicenter study was to quantify the proportion of false-positive CHD diagnoses made during first-trimester fetal heart evaluations conducted by trained operators. This study seeks to provide valuable insights into the reliability of early CHD detection and the implications of false positives in prenatal care.

## 2. Materials and Methods

This multicenter retrospective study involved all pregnant women undergoing first-trimester combined screening for aneuploidies in six Fetal Medicine units across Italy between January and October 2022. A database search was performed to collect all first-trimester screening performed in the study period. Among these, only the cases with suspicion of CHDs at the first-trimester scan were included for analysis. The first-trimester screening for fetal aneuploidies was performed according to the Fetal Medicine Foundation (FMF) criteria [22]. Singleton and twin pregnancies were included. In all cases, an ultrasound evaluation of the fetal heart was performed by fetal medicine specialists with extensive experience in first-trimester scanning, following an extended anatomical protocol [20]. The following cardiac views were obtained using 2-dimensional (2-D) sonography with gray-scale imaging and/or color Doppler: an axial view of the fetal abdomen and thorax to assess visceral situs; the four-chamber view (4CV); and the three-vessel and trachea view (3VTV). Transvaginal ultrasounds were performed in cases where transabdominal scans provided suboptimal views. Customized settings were used for the first-trimester fetal heart scans, minimizing the ultrasound exposure of the fetuses, especially in Doppler mode. In cases identified as high-risk for aneuploidies, with nuchal translucency (NT) thicknesses at or above the 99th percentile, or in the presence of major fetal structural defects, chorionic villus sampling (CVS) or amniocentesis was offered. In all suspected CHD cases, FE performed by a fetal cardiologist was offered by 16 weeks and/or by 22 weeks of pregnancy. When a CHD was diagnosed with the FE, a follow-up scan was offered to each patient. All live-born neonates with cardiac anomalies underwent echocardiography within the first week of life. All cases not assessed by a fetal or pediatric cardiologist, either prenatally or postnatally, or by a postmortem examination were excluded from this analysis.

## 3. Results

In the study period, a total number of 4300 fetuses were evaluated in the first trimester. Forty-six (1%) CHDs were suspected between 11^+0^ and 13^+6^ weeks of gestation, with 44 cases from singleton pregnancies and 2 cases from twin pregnancies where only one twin was suspected of having a CHD. Within this group, the median maternal age was 35.5 years (IQR 25–4), the median gestational age was 12^+4^ weeks (IQR 11+3–14+0), and the median BMI was 23.1 (17–40). The gestational age at the time of examination was between 11^+0^ and 11^+6^ weeks in 7 cases (15.2%), between 12^+0^ and 12^+6^ weeks in 25 cases (54.3%), and between 13^+0^ and 13^+6^ weeks in 14 cases (30.4%) (Table 1).

Among the 46 suspected CHDs, 24 cases were excluded, as early termination of pregnancy was opted for and postmortem examination was not available for diagnostic confirmation. For the remaining 22 cases, the average NT was 2.5 mm [IQR 1.1–13.6], among which 8 fetuses (36%) had NT values above the 99th percentile and 12 fetuses (54%) had NT values between the 95th and the 99th percentiles. The combined first-trimester risk for aneuploidies was increased in 13 cases (65%). Additional extracardiac anomalies were present in 11 fetuses (50%). Twenty patients opted for invasive procedures (90%) (Table 2). Due to the suspicion of CHDs, all women underwent fetal karyotype and chromosomal microarray analysis (CMA). In 10 cases (50%), genetic abnormalities were diagnosed, including trisomy 21 (4 cases), trisomy 18 (1 case), monosomy X (2 cases), triple X syndrome (1 case), 46XXmicrodup16 (1 case), and 46XXmicrodel16 (1 case). In 11 cases (50%), both the 4CV and the 3VTV were reported as abnormal; in 7 cases (32%), the 4CV was abnormal and the 3VTV was normal; and in 4 cases (18%), the 3VTV was abnormal and the 4CV was normal (Table 2). Detailed characteristics of the included cases are given in Appendix A.

The most common CHDs suspected were AVSDs and hypoplastic left-heart syndrome (HLHS) (Figure 1 and Figure 2 and Appendix A).

Fetal echocardiography was performed by the 16th week of gestation in 14 cases (64%) and after 16 weeks in 8 cases (36%). In 19 cases (86%), the fetal cardiologist identified the presence of CHDs, which was confirmed at postnatal echocardiography, surgery, or postmortem examination in all cases. In 16 out of these 19 cases, parents opted for termination of the pregnancies. The presence of CHDs was ruled out in 3 out of 22 cases (13%) within 16 weeks’ gestation, and normal cardiac anatomy was confirmed in the second-trimester FE and postnatally. The anomalies suspected in the first trimester and not confirmed were an abnormal aspect of the 4CV, suggestive of an atrioventricular septal defect (AVSD) in one case and an abnormal aspect of the 3VTV in two cases (Figure 3).

Invasive tests were performed in 2 out of these 3 cases. In one case, it was performed because of an increased risk of aneuploidies associated with the suspicion of an AVSD, and in the second case, it was performed because of an NT between the 95th and 99th percentiles with an abnormal 3VTV. In both cases, the genetic workup results were normal.

## 4. Discussion

The anatomical evaluation of the fetal heart typically takes place from 18 weeks’ gestation onward, as this period allows for clearer and more detailed imaging of cardiac structures due to the larger fetal size and further development of the cardiovascular system [19,23]. By this stage, the chambers, valves, and vessels of the heart are generally well-defined and can be assessed with higher accuracy, providing a better foundation for diagnosing CHDs. Early detection of CHDs is crucial, as it enables families and healthcare providers to make informed decisions about prenatal care, delivery planning, and immediate postnatal interventions if needed. Early awareness of a potential CHD can also provide valuable time for parents to prepare emotionally and logistically for specialized care after birth. Historically, first-trimester screening for CHDs has focused on identifying indirect markers rather than attempting to visualize the heart’s structures directly. Some of the key markers assessed include increased NT, tricuspid regurgitation, and altered blood flow in the ductus venosus [24]. These markers are useful because they can be indicators of broader chromosomal abnormalities or structural defects, including CHDs. However, these indirect signs only hint at potential anomalies rather than providing a clear visualization of the heart itself, which limits their diagnostic precision. For instance, while an increased NT measurement may suggest a higher risk for heart defects, it can also be associated with other non-cardiac conditions, leading to potential false positives and further diagnostic uncertainty. In recent years, improvements in ultrasound technology and the increasing skill levels of trained operators have enabled more direct and detailed assessments of the fetal heart earlier in pregnancy. Specifically, direct visualization of the 4CV and the 3VTV during the first trimester has proven to be a more accurate method for early CHD detection [16,25]. The 4CV allows clinicians to examine the symmetry and structure of the heart’s four chambers, while the 3VTV provides an important overview of the connections between the major blood vessels and the heart, including the pulmonary artery and aorta. These views are particularly informative in identifying major structural anomalies that might impact blood flow and cardiac function. Studies have demonstrated that incorporating these direct views into first-trimester screening protocols yields higher accuracy in detecting CHDs compared to reliance on indirect markers alone [18,25,26]. When combined with advanced ultrasound equipment capable of capturing high-resolution images and color Doppler technology, which helps visualize blood flow, these views allow operators to assess the fetal heart in greater detail. Additionally, increased training and experience among ultrasound technicians and fetal medicine specialists have further enhanced the effectiveness of these early assessments. As a result, first-trimester fetal heart evaluation has become a more reliable option in the early detection of CHDs, allowing for quicker diagnosis and intervention planning. As a matter of fact, recent guidelines recommend including the direct evaluation of the fetal heart in the first trimester through the 4CV and 3VTV with or without color Doppler in high- or low-risk populations [20,21,27]. This recommendation reflects a growing recognition of the benefits of early detection, particularly given that technological advancements and operator expertise have significantly improved the accuracy of early FE. The rationale behind this approach lies in the accessibility of a reliable diagnostic test—early FE—that is feasible in the first or early second trimester and allows for the timely confirmation or exclusion of suspected CHDs during early gestation. Detecting CHDs in the first trimester is increasingly prioritized because early suspicion or diagnosis of cardiac anomalies can lead to further investigation for associated conditions. CHDs frequently coexist with genetic and/or extracardiac structural anomalies, as corroborated by our study’s findings: nearly half of the detected cases also presented with either genetic abnormalities or additional extracardiac defects. This association underscores the importance of early CHD screening, as it may alert healthcare providers to a broader spectrum of potential fetal health concerns, prompting them to recommend invasive prenatal testing when a high risk of chromosomal abnormalities is identified. Therefore, the early suspicion or detection of CHDs may anticipate the diagnosis of chromosomal abnormalities, constituting an indication for invasive prenatal testing rather than screening with cell-free DNA [28]. Early identification is, therefore, essential to plan the most appropriate diagnostic workup. However, the evaluation of the fetal heart in the first trimester largely depends on operator experience, the time allocated for the examination, and ultrasound machine settings, which should be optimized for this application [29,30]. These aspects, combined with technical challenges due to the small size of the cardiac structures, increase the likelihood of misinterpreting ultrasonographic findings, thereby raising the false-positive and -negative rates. In our study, the first-trimester assessment of the fetal heart was performed by trained operators with at least three to five years’ experience in obstetric screening ultrasound and specifically in first-trimester evaluation of fetal anatomy. However, 13% of cases were suspected to have CHDs not confirmed in subsequent scans. The false-positive rate in first-trimester scans is challenging to determine due to the early surgical termination that precludes postmortem confirmation [18]. As this lack of diagnostic confirmation might lead to an underestimation of true positive cases, we decided to exclude all cases where diagnostic confirmation was missing. By doing so, our results indicate that false positives may be more common than previously reported. False-positive results can impact pregnancy outcomes by leading to unnecessary additional tests, causing stress, and increasing financial costs for the healthcare system. In our case series, termination of pregnancy was opted when CHDs were confirmed in FE, indicating that first-trimester suspicion of CHDs did not, per se, significantly influence the pregnancy outcome. However, great caution should be taken following the early identification of a CHD, as the negative impact of a false-positive diagnosis cannot be underestimated. On the other hand, research has shown that while women who receive false-positive diagnoses may experience temporary increases in anxiety, these levels typically return to normal once the defects are ruled out in subsequent scans, and this does not impact the overall pregnancy experience [31]. Additionally, most women express a preference for early fetal anatomy studies, as they feel reassured by normal findings or appreciate having more time to make decisions if positive findings occur [31].

In instances where CHDs are suspected during first-trimester screening, it is essential to communicate openly with the expecting parents regarding the limitations of early fetal heart assessments. Early examination provides valuable initial insights but is not definitive, as the small size and rapid development of the fetal heart make early assessments more challenging. Parents should be informed that while early screening can raise flags for potential issues, it is not a conclusive diagnostic evaluation, and further assessments in the second trimester are necessary to confirm or rule out CHDs with greater accuracy. Highlighting the role of first-trimester evaluation as a preliminary screening tool—rather than a diagnostic procedure—can help manage parental expectations and reduce the emotional impact associated with preliminary findings that might be inconclusive or suspicious. Given this context, it is both reasonable and expected for early screenings to occasionally yield findings that are indeterminate or suggestive of CHDs, thus necessitating follow-up FE at a later gestational age for a more precise evaluation. The importance of follow-up scans lies in the ability to more accurately assess fetal heart anatomy, reducing the likelihood of false positives and enabling healthcare providers to offer more targeted counseling and intervention options if needed. This study has certain limitations, primarily stemming from its retrospective design, which may have introduced biases related to data collection and analysis. Additionally, the absence of data from screen-negative populations (fetuses initially assessed as not having CHDs) limits the ability to calculate accurate false-positive and false-negative rates for CHD detection during first-trimester screening. These rates are crucial for understanding the overall effectiveness and accuracy of the screening process but were beyond the primary objective of this study. The focus here was on assessing the performance of first-trimester CHD screening in terms of identifying false-positive cases, thus providing insight into the reliability and limitations of early cardiac evaluation.

## 5. Conclusions

The main finding of this study underscores that first-trimester evaluation of the fetal heart, while valuable for early detection, is associated with a higher-than-expected rate of false-positive diagnoses of CHDs. This elevated risk of false positives highlights the limitations inherent in early screening due to the small size of the fetal heart and the developmental changes that occur as pregnancy progresses. However, despite the occurrence of false positives, these initial findings typically do not influence the overall outcome of the pregnancy. During prenatal counseling, it is essential to communicate to parents that a suspected abnormality in a first-trimester ultrasound does not definitively indicate the presence of a CHD. Instead, it should be viewed as an initial indicator that warrants further investigation. Parents should be reassured that follow-up FE is always necessary to confirm or exclude a CHD diagnosis, providing a more accurate assessment of the fetal heart anatomy and prognosis. By emphasizing that first-trimester evaluation serves as a preliminary screening tool rather than a conclusive diagnostic procedure, healthcare providers can help manage parental expectations and alleviate undue anxiety. The importance of follow-up scans cannot be overstated, as they allow for a more detailed and reliable examination of the fetal heart, ultimately guiding the most appropriate prenatal and postnatal care.

## Figures and Tables

**Figure 1 diagnostics-14-02543-f001:**
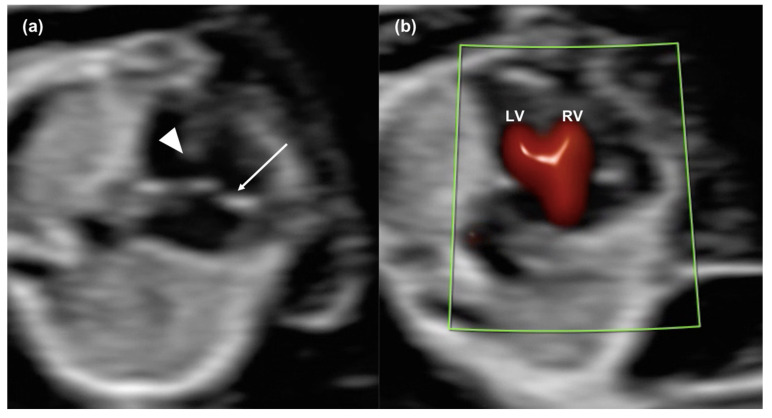
A case of a common atrioventricular valve at 12 weeks, assessed in gray-scale and with color Doppler. In (**a**), the 2D image shows the common atrioventricular valve (arrow) and the wide ventricular septal defect (arrowhead). In (**b**), the color Doppler clearly shows the single atrioventricular inlet entering two separated ventricles. RV; right ventricle; LV, left ventricle.

**Figure 2 diagnostics-14-02543-f002:**
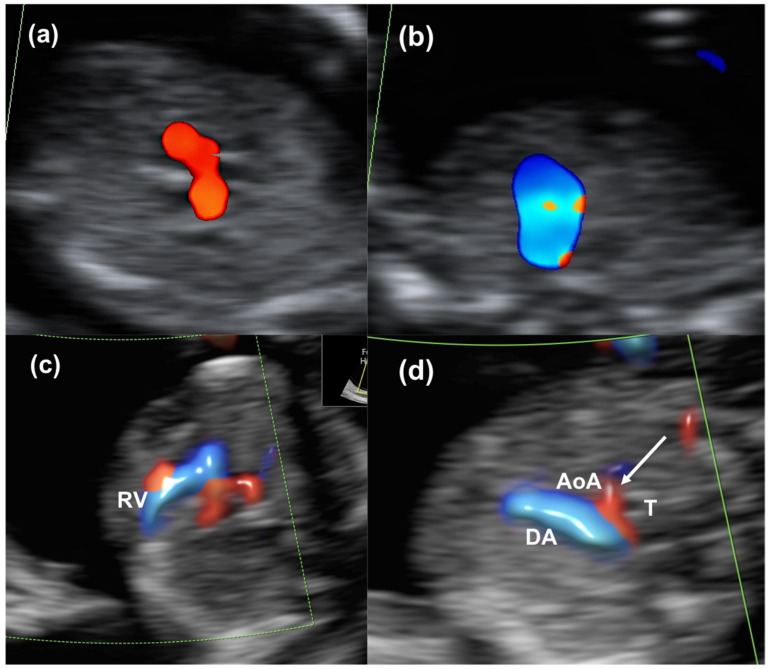
A case of hypoplastic left-heart syndrome at 12 weeks, assessed with color Doppler evaluation. In (**a**,**b**), images obtained with a transabdominal approach show in (**a**) the filling of one ventricle in the four-chamber view and in (**b**) the presence of a single vessel in the three-vessel trachea view. In (**c**,**d**), the same case obtained with a transvaginal approach shows in (**c**) the filling of the right ventricle (RV) in the four-chamber view and in (**d**) the presence of a small aortic arch with reverse flow (white arrow). Ductal arch (DA); aortic arch (AoA); trachea (T).

**Figure 3 diagnostics-14-02543-f003:**
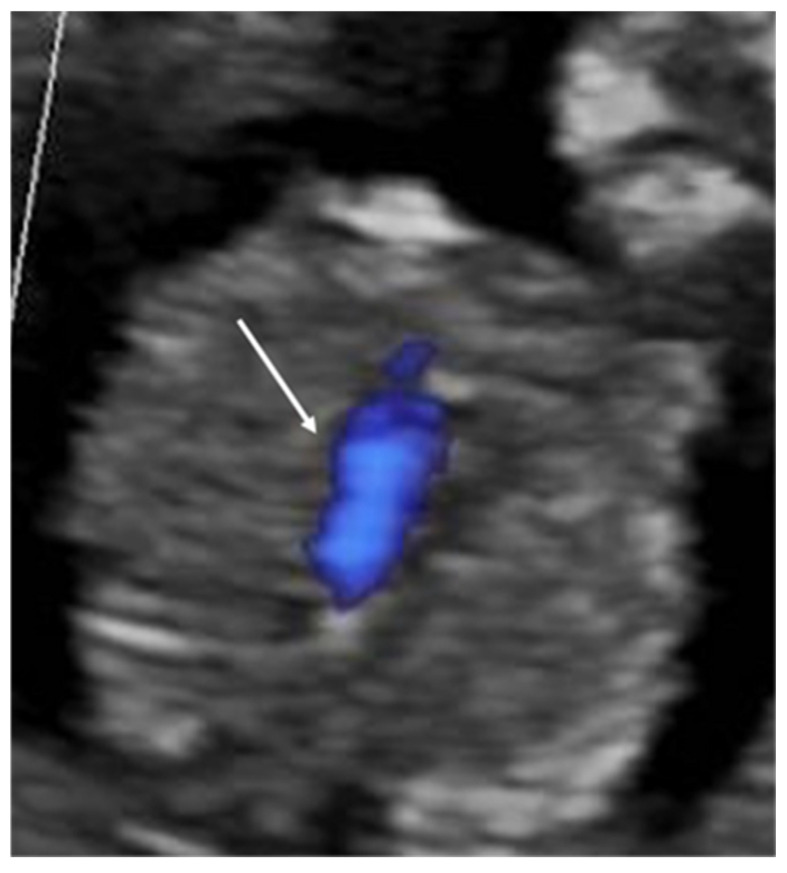
A false-positive case showing the presence of one single vessel (white arrow) at the three-vessel and trachea view with color flow mapping.

**Table 1 diagnostics-14-02543-t001:** Maternal characteristics of the 46 cases with suspected CHDs in the first trimester.

Maternal Characteristics	Median (IQR)
Maternal age (years)	35.5 (25–47)
Body mass index (kg/m^2^)	23 (17–40)
Gestational age (weeks)	12 + 4 (11 + 3–14 + 0)
	**Number (*n*, %)**
Pregnancy	
Singleton	44 (96%)
Multiple	2 (4%)
Mode of conception	
Spontaneous	41 (89%)
IVF	2 (4%)
Smoking	4 (9%)
Family history of cardiac defects	1 (2%)
Maternal diabetes	1 (2%)

IQR, interquartile range; IVF, in vitro fertilization. Data are given as medians (IQR), *n* (%).

**Table 2 diagnostics-14-02543-t002:** Fetal characteristics of the included cases with suspected CHDs in the first trimester.

Fetal Characteristics	CHDs (22)
Nuchal translucency (mm)	**Median (IQR)** 2.5 (1.1–13.6)
	**Number (*n*, %)**
Nuchal translucency > 99th percentile	8 (36%)
Nuchal translucency > 95th percentile	12 (54%)
First-trimester high risk for fetal aneuploidies	13 (65%)
Associated fetal abnormalities	11 (50%)
Invasive procedures (CVS or amnio)	20 (91%)
Genetic abnormalities	10 (50%)
Abnormal 4CV	7 (32%)
Abnormal 3VTV	4 (18%)
Abnormal 4CV and 3VTV	11 (32%)

CHDs, congenital heart defects; IQR, interquartile range; CVS, chorionic villous sampling; amnio, amniocentesis; 4CV, four-chamber view; 3VTV, three-vessel and trachea view. Data are given as medians (IQR), *n* (%).

## Data Availability

The data presented in this study are available on request from the corresponding author due to the privacy policies of the hospitals involved.

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
