# Peer review of "False-Positive Diagnosis of Congenital Heart Defects at First-Trimester Ultrasound: An Italian Multicentric Study"

_diagnostics, 2024, doi:10.3390/diagnostics14222543_

Round 1
Reviewer 1 Report
Comments and Suggestions for Authors
False positive diagnosis of congenital heart defects at first-trimester ultrasound: an Italian multicentric study. – Diagnostics
Abstract – the authors list objectives, when it appears that there is only 1 objective. To clarify.
Who are the experienced operators – to clarify. The authors refer to experience, however, did these operators receive any formal training?
Results to reword the following statement - Results. ‘Among 4.300 fetuses’ to for example – from a population sample of ….
Introduction.
A brief introduction to the area being studied is given.
At the end of this section, the authors now refer to ‘the aim of the study’ when previously this was listed as an objective/s. It appears that the authors may be using the terms interchangeably and/or are not able to distinguish between the two. Suggest that this issue is clarified on resubmission of the text.
Materials and methods
Statements, such as – we included – should be reworded and written in more academic English.
How were the patients recruited? Was any ethical permission sought and obtained? Did the participants consent to participate? The patients were followed through until the baby was born or the pregnancy terminated – how was this done? Was this done in accordance with GDPR regulations since the study took place in an EU member state? The issue of ethics is only addressed as an additional note at the end of the text. Suggest that this is included under methodology with an explanation of procedures in the centres from where the data was collected.
Did the patient results also include the name of the person who carried out the scans? How did the authors know who performed the scans? Was any analysis done linked to the person who carried out the scan?
Results
These are well presented.
Discussion
Training and experience – the authors keep referring to experience. Did they consider what training, if any, they had received regarding the safe use of the ultrasound devices as well any qualifications in the field of ultrasound scanning? This is a short coming of the study.
Expert operators
Again, to review writing
The discussion is rather limited. This may be reviewed to make it more critical.
Comments on the Quality of English LanguageIn some places the authors need to review the writing and use more academic English.
English (US) spelling is applied in the text.
Reviewer 2 Report
Comments and Suggestions for Authors
Dear Editor and Authors,
I read the paper entitled ‘False positive diagnosis of congenital heart defects at first-tri- 2
mester ultrasound: an Italian multicentric study' with great interest.
The title describes the core message of the paper.
The abstract incorporates key messages, in a concise manner.
The structure of the paper is accurate.
However, I have same suggestions regarding this paper, which should be taken into account in future research.
1. The studied group is relatively small, after exclusion criteria.
2. There is no statistical methods incorporated into this study. Only numbers and percentages of different parameters are presented.
Reviewer 3 Report
Comments and Suggestions for Authors
The article is devoted to the analysis of false positive diagnoses of congenital heart defects in ultrasound examination in the first trimester of pregnancy in Italian women. It draws our attention to the fact that ultrasound examination of the fetal heart in the first trimester of pregnancy is associated with technical problems associated with the small size of the heart structures, largely depends on the experience of the operator, the time allocated for the examination, and the optimal settings of the ultrasound machine. These factors increase the frequency of false positive and negative results. Despite the fact that in this study, patients underwent expert ultrasound examination, in 13% of cases the diagnosis was false-positive. Preliminary analysis of congenital heart defect in 24 cases out of 46 led to premature termination of pregnancy. In those cases where the pregnancy was preserved, the diagnosis was not confirmed in 13%. In this regard, the authors conclude that in cases where a fetus is suspected of having a congenital heart defect during first trimester screening, it is extremely important to inform the couple about the limitations of early fetal examination and to emphasize the importance of subsequent ultrasound examinations. For these reasons, the authors recommend that the examination of cardiac anatomy in the first trimester of pregnancy be considered a screening tool rather than a diagnostic method. The study is scientifically novel and adds to our knowledge of prenatal screening for congenital heart defects. It provides practical recommendations for counseling couples who are faced with a suspicion of a fetus having a heart defect and has great practical value. The material and methods are described correctly. The discussion is comprehensive, the conclusions are justified. The article indicates the limitations of the study. The article can be recommended for publication.
Round 2
Reviewer 2 Report
Comments and Suggestions for Authors
Manuscript was improved sufficiently.